# Identification of CAND1 as a DNA-dependent protein kinase-regulated coactivator of androgen receptor and the ARv7 splice variant

**Ross A. Hamilton**[1], **Basil Paul**[1], **Ping Yi**[1,2], **Kimal Rajapakshe**[1], **Anil K. Panigrahi**[1], **Sandra L. Grimm**[1,3], **Cristian Coarfa**[1,3], **Anna Malovannaya**[1,3,4,5], **Nancy L. Weigel**[1,3], **David M. Lonard**[1,3]*, **Charles E. Foulds**[3,6,7]*

1 Department of Molecular and Cellular Biology, Baylor College of Medicine, Houston, Texas, United States of America, 2 Department of Biology and Biochemistry, Center for Nuclear Receptors and Cell Signaling, University of Houston, Houston, Texas, United States of America 3 Dan L. Duncan Comprehensive Cancer Center, Baylor College of Medicine, Houston, Texas, United States of America, 4 Department of Biochemistry and Molecular Pharmacology, Baylor College of Medicine, Houston, Texas, United States of America, 5 Mass Spectrometry Proteomics Core, Advanced Technology Cores, Baylor College of Medicine, Houston, Texas, United States of America, 6 Lester and Sue Smith Breast Center, Baylor College of Medicine, Houston, Texas, United States of America, 7 Department of Medicine, Baylor College of Medicine, Houston, Texas, United States of America,

* dlonard@bcm.edu (DML); foulds@bcm.edu (CEF)

## Abstract

ARv7, the most prevalent androgen receptor (AR) variant in castration-resistant prostate cancer, lacks the ligand binding domain (LBD), rendering it resistant to LBD-targeted therapies. Identifying new therapeutic targets requires defining the coregulators and associated regulatory enzymes that govern AR and ARv7 transcriptional activity. Here, we have developed a cell-free DNA pulldown assay employing androgen response elements (AREs) to isolate and characterize the AR- and ARv7-associated coregulator complexes formed on DNA. Mass spectrometry analyses of ARE DNA pulldowns revealed previously unrecognized AR and ARv7 associating coregulators, such as cullin-associated NEDD8-dissociated protein 1 (CAND1), in addition to previously known coregulators. ARv7 showed enhanced recruitment of a subset of AR associating coregulators. Knockdown of CAND1 in prostate cancer cells reduced the expression of AR and ARv7 target genes, supporting its role as a coactivator. Bioinformatic analyses of human prostate cancer clinical datasets revealed that *CAND1* mRNA level correlated with disease status, with higher expression correlated with metastatic prostate cancer and poorer patient survival. We further show that DNA-dependent protein kinase (DNA-PK) phosphorylates both AR and ARv7, enhances their transcriptional activities, and stabilizes the interaction of CAND1 with AR- and ARv7- coregulator complexes. Collectively, these findings suggest that DNA-PK stimulates the AR and ARv7 activity through its enzymatic function and by stabilizing (or reinforcing) coactivator interactions, including those involving CAND1.

**Data availability statement:** All relevant data are within the paper and its Supporting Information files. The raw MS proteomics data for the the (-/+) ARv7 ARE pulldowns have been deposited to the ProteomeXchange Consortium via the PRIDE partner repository with the dataset identifier PXD021462. These raw data are now publicly available.

**Funding:** The study was funded by Eunice Kennedy Shriver National Institute of Child Health and Human Development grants R01HD008188 and R01HD007857 (D.M.L); Cancer Prevention and Research Institute of Texas grant RP150648 (N.L.W); U.S. Department of Defense grants W81XWH-17-1-0236 (N.L.W) and W81XWH-21-1-0404 (P.Y); National Cancer Institute grants P30 CA125123 (A.M) and U54CA274321 (C.C); and National Institute of Environmental Health Sciences grant P42 ES027725 (C.C). The above funders had no role in study design, data collection and analysis, decision to publish, or preparation of the manuscript. CoRegen Inc. provided support in the form of salaries for authors [A.K.P, P.Y, D.M.L, and C.E.F], but did not have any additional role in the study design, data collection and analysis, decision to publish, or preparation of the manuscript. The specific roles of these authors are articulated in the 'author contributions' section. There was no additional external funding received for this study.

**Competing interests:** C.E.F and D.M.L disclose other support from CoRegen Inc. unrelated to the current study.

In sum, this work advances our understanding of AR isoform actions and identifies additional potential therapeutic targets for castration-resistant prostate cancer.

## Introduction

Prostate cancer is the most common non-cutaneous cancer in American men, and the second most common cause of cancer death in U.S. males [1]. Because of the importance of the androgen receptor (AR), a hormone-activated transcription factor (TF), in the growth of both normal prostate and prostate cancer, androgen deprivation therapies (ADTs) and androgen receptor signaling inhibitors (ARSI) are used to treat prostate cancer patients with metastatic prostate cancer [2–4]. Initially, tumors respond to ADT, but resistance typically develops within two years resulting in castration-resistant prostate cancer (CRPC) [5–7].

Despite the resistance to ADT/ARSI, most CRPC remain dependent on AR [3] and there are no established cures. The mechanisms behind aberrant AR reactivation contributing to CRPC include specific AR mutations, AR overexpression/amplification, *de novo* steroidogenesis, altered expression of coregulators (CoRs), AR hypersensitivity to androgens and expression of constitutively active AR variants [8–15]. The AR variants lacking the ligand binding domain (LBD) are expressed in some CRPC as a result of alternative splicing events and/or genomic rearrangements [12,15–17]. The most prevalent and well characterized AR splice variant is ARv7, which contains the amino (N)-terminus and DNA binding domain as well as a small strech of unique amino acid sequence [16]. Expression of ARv7 correlates with resistance to second generation ADT/ARSI therapies such as abiraterone/enzalutamide [18,19] suggesting a key role for variants in resistance to ADT. The lack of an LBD means that other portions of the molecule or its downstream activities must be targeted to block its activity. Unlike estrogen receptor alpha (ERα) whose major transactivation domain and site for interactions with coactivators is the LBD [20,21], the major transactivation domain in AR is in the N-terminus and both functional studies [22] and recent cryoEM structural studies [23] show that the p160 steroid receptor coactivator family members interact with the N-terminus of AR.

Nuclear receptors act as TFs by recruiting a variety of CoR proteins with enzymatic activities to specific DNA binding sites (reviewed in [24,25]). Proteins with a variety of enzymatic activities have been identified as interacting with AR and variants and playing a role in its transcriptional activity (reviewed in [24–28]); however, identification of additional proteins, many of which interact transiently or in complexes and often are low abundance proteins, is incomplete. Thus far, approaches including co-immunoprecipation (Co-IP) with immunoblotting and/or mass spectrometry (MS) [29–33] and 'proximity mapping' using *in vivo* biotinylation of AR or ARv7 and their pulldown [34,35] have been used to identify the AR/ARv7 interactors. However, the interactions of AR/ARv7 with their coactivators on DNA may be very different from those in solution, and thus methods such as rapid IP MS of endogenous proteins (RIME) assays [36] or chromatin immunoprecipitation (ChIP) assays (e.g., [37,38]) have been performed to identify AR-associated CoRs in an unbiased or targeted

fashion, respectively. One limitation of these assays is that a large variety of complexes are pulled down, not only the protein complexes bound to AR or AR variants, but those bound to other TFs linked to the same DNA fragment. Since AR and ARv7 bind to common sites, but also to unique sites in the DNA [39], this approach is not optimal for identifying differences in protein binding in AR versus ARv7 complexes.

To identify proteins that interact with AR and ARv7, we took advantage of some of our previous studies of estrogen receptor alpha (ERα). Previously, we developed estrogen response element (ERE) DNA pulldown assays combined with unbiased liquid chromatography (LC)-MS to identify proteins associated with ERα and to determine the contribution of proteins to the transcriptional activity of ERα [40]. Among the interesting findings from the ERα study was the observation that DNA-dependent protein kinase (DNA-PK) enhanced the transcriptional activity of ERα by dynamically modulating the composition of the ERα complexes via phosphorylation of ERα and some coregulators [40].

DNA-PK also has been reported to associate with AR in the absence of DNA and to stimulate AR target genes such as *PSA* and *TMPRSS2* in prostate cancer cells [41], revealing it as a coactivator. More recently using a proximity-biotinylation approach, DNA-PK was shown to associate with ARv7 and DNA-PK knockdown/inhibition reduced expression of its target genes, including *PSA*, *KLK2*, and *UBE2C*, suggesting a coactivator role [35]. However, mechanistically how DNA-PK functions as a coactivator of AR/ARv7 was not examined.

In this study, we employed biochemical approaches to investigate the proteomics of ARv7-CoR complexes and compared them to isolated AR-CoR complexes. We used a DNA fragment containing three AREs, in order to directly compare AR and ARv7 capacity to recruit coregulators and regulate transcription. This study identified a new ARv7 coactivator (CAND1) that appears to have a higher affinity for ARv7 as compared to AR and uncovered a role for DNA-PK in regulation of the stability of AR/ARv7-ARE complexes.

## Materials and methods

### Reagents

All chemicals used to make buffers for preparing nuclear extracts, purifying FLAG-tagged AR (M2 beads and 3xFLAG peptide), inducing expression of ARv7 (doxycycline), cell-free transcription, and performing ChIP assays were from Sigma-Aldrich. R1881 was purchased from Perkin Elmer. Protease inhibitors, AMP-PNP, ECL Plus, and UniversalLibrary Probes for qPCR were purchased from Roche. PCR reagents, SuperScript II Reverse Transcriptase, Dynabeads M280 streptavidin, and NuPage gels were purchased from Invitrogen. SDS sample buffer and Protein A/G Magnetic Dynabeads were purchased from Pierce. Mini-PROTEAN TGX Precast gels were purchased from Bio-Rad. 1x kinase buffer and ATP were purchased from Cell Signaling Technology. DNA-PK inhibitor NU7441 was purchased from Tocris Bioscience. λ protein phosphatase (LPP) was purchased from New England Biolabs. TransIT-TKO was purchased from Mirus. Tri-reagent was purchased from Molecular Research Center, Inc. For qPCR, Taqman Universal master mix II and SYBR green mix were purchased from Applied Biosystems. PCR purification kit was purchased from Qiagen. Sources and usage conditions for antibodies used in this study are given in S1 Table in S1 File.

### Cell lines

LNCaP cells (RRID:CVCL_0395) were obtained from the BCM tissue culture core (originally from the ATCC as catalog #CRL-1740), while LNCaP-doxycycline (dox)-inducible ARv7 cells (LNCaP^pHageAR--V7; RRID:CVCL_A1BM) were described previously [42]. Short tandem repeat profiling (STR) validated LNCaP and LNCaP-dox inducible ARv7 cell line authenticity (done at MD Anderson, January of 2019). The cell lines were cultured for no longer then 3 months after a thaw of the STR validated stock. All lines were tested for mycoplasma every 3 months and tested negative. LNCaP lines were maintained in RPMI 1640 media (Invitrogen) (supplemented with 10% fetal calf serum and penicillin/streptomycin) and switched to hormone-depleted media (10% charcoal stripped fetal calf serum) before treatments. The LNCaP^pHageAR--V7 cells were carried in medium containing G418 to maintain selection pressure, but G418 was not included in experimental plates.

## Nuclear extract preparation

The nuclear extract (NE) preparation from HeLa S3 cells (purchased from National Cell Culture Center) was conducted following the standard Dignam protocol [43], while extraction of LNCaP cells was done according to our published method for MCF-7 NE preparation [40]. Bradford assays (Bio-Rad) were utilized to determine protein concentration and aliquots were snap-frozen in liquid nitrogen and stored at -80°C until usage.

## Recombinant AR/ARv7 purification

Recombinant AR protein was purified using *Sf9* insect cells that were infected with a Flag epitope-tagged AR expressing baculovirus (courtesy of the BCM Monoclonal Antibody/Recombinant Protein Expression Core Facility). Cells were cultured in the presence of 1 µM R1881 for 24 hours post-infection and then harvested 24 hours later. Cells were then washed and centrifuged at 4°C for 10 min at 5,000 rpm. Purification of AR from cell pellets was based on [44], with some modifications as listed. Cell pellet was then resuspended in 50 mL hypotonic buffer with 1 µM R1881 (not DHT) and complete mini protease inhibitors (Roche) and not Protease Inhibitor Cocktail Set III. Nuclei were then resuspended in 12 mL of nuclear extraction buffer with 1 µM R1881, without β-glycerophosphate and NaF, and complete mini protease inhibitors and extracted as described [44]. Extract was then clarified by centrifugation at max speed (12,000 rpm, not 50,000xg) for 30 min and supernatant containing protein was transferred to a clean chilled tube. M2 FLAG beads, not Streptavidin Mutein Matrix gel, were washed 3 times in equilibration buffer (10 mM Tris-HCl (pH 8.0), 0.3 M NaCl). Beads were resuspended in the 12 mL nuclear extract and incubated overnight at 4°C with gentle mixing by rotation. After this, beads with bound AR were gently spun down at 500xg and the supernatant was removed. Beads were then resuspended with 1.25 ml wash buffer containing 1 µM R1881 and transferred into 1.5 mL Eppendorf tubes. Washes were done twice. Flag-tagged AR protein was eluted from the beads using 250 µl elution buffer without biotin containing 1 µM R1881 and 3xFLAG peptide (0.4 µg/mL). Beads were incubated on ice for 10 min and 4 elutions were repeated. Elutions were pooled and flash frozen in liquid nitrogen. Protein concentration was determined by Coomassie blue staining versus known amounts of bovine serum albumin (BSA). Purified ARv7 protein, provided by Dr. Ping Yi, was prepared using a similar procedure except that R1881 was not included.

## 3xARE-E4 DNA fragment synthesis

Three copies of a canonical ARE from the rat *TAT* gene was cloned into pIE-0 as described previously (called pPRE3-E4) [45]; this construct has a minimal TATA box from the *Adenovirus E4* gene downstream of the 3xAREs. A 853 bp 3xARE-E4 doubly biotinylated fragment was made from this construct by PCR using *Taq* DNA polymerase and E4BioF/E4BioR primers biotinylated at their 5' ends (sequences shown in S2 Table in S1 File).

## Androgen response element (ARE) pulldown assay with and without ATP treatment (ATP for kinase activation)

30 µl M280 streptavidin Dynabeads were prepared for biotinylated DNA binding as in [40]. Beads were then resuspended in 150 µl D-PBS and incubated with 2 µg of biotinylated 3xARE-E4 DNA for 1 hour at 4°C. DNA-bound beads were washed as described above. Two µg recombinant purified AR or 0.25 µg purified ARv7 (or listed amounts in figure legends) or its storage buffer were then added along with the appropriate ligand (100 nM R1881 or 0.1% ethanol as vehicle control) first to the 3XAREs beads for 10 minutes at room temperature in a final volume of 150 µl supplemented with Buffer D (20 mM HEPES, pH 7.9, 10% glycerol, 0.1 M KCl, 0.2 mM EDTA, 0.5 mM PMSF, and 0.5 mM DTT) to pre-bind the AR proteins for use in HeLa NE ARE pulldowns. Beads were pelleted and all supernatant containing unbound AR protein was removed. The ARE DNA beads were then resuspended with 1 mg clarified (by centrifugation at 12,000 rpm for 15 min at 4°C) HeLa NE or LNCaP NE. Additional EDTA/EGTA was added for a final concentration of 1 mM. Reactions were incubated with rotation at room temperature for 45 min. Beads were washed twice using ice-cold NETN buffer (20 mM Tris-Cl,

pH7.5, 150 mM NaCl, 1 mM EDTA, 0.5% Igepal CA-630), followed by one wash in ice-cold D-PBS. PBS was removed and beads were resuspended in 20–30 µl 2x SDS sample buffer to be boiled. Protein samples were loaded on 4–15% Mini-PROTEAN TGX™ Precast gels for immunoblotting (30–50% of final beads) or on NuPage gels (8% Bis-Tris in MOPS buffer) for MS (for preparative separation-gel slice excision) (90% of final beads). ATP treatment (to activate associated kinases) of AR/ARv7 CoR complexes (i.e., *in vitro* phosphorylation) involved resuspending the washed complexes in 1x kinase buffer containing 0.5 mM ATP and incubating them at 30°C for up to 30 min. Where appropriate, AMP-PNP was added in place of ATP or NU7441 was pre-incubated for 5 minutes before adding ATP. λ protein phosphatase (LPP) or LPP Buffer was added 5 minutes post-ATP (30 min) treatment addition. After incubation, the beads were pelleted by magnet, the supernatant was removed, and the beads were resuspended in 20–30 µl 2x SDS sample buffer for subsequent immunoblotting analysis (30–50% of final beads).

## Cell-free transcription assay

These assays (run in duplicate) utilized the above 3xARE-E4 template, except it was assembled into chromatin using HeLa core histones and a salt dilution method that we previously described [46]. Briefly, the chromatized 3xARE-E4 DNA template (0.2 pmoles) was incubated with 50 µg HeLa cell NE, with and without purified AR with 100 nM R1881 or ARv7 (the amount of AR or ARv7 added to reactions is indicated in respective figures), 1 mM ATP, 0.9 mM acetyl CoA, 2% poly(vinyl alcohol), Hepes buffer and salts as defined in [23], in a 50 µl reaction at room temperature for 25 min. 5 µL of 5 mM NTPs were then added to allow for active transcription and the samples were transferred to 30°C for 50 min. RNA was isolated by Tri-reagent and further treated with Turbo DNase (Ambion) to remove any remaining DNA template. Two µL of each purified RNA sample (5% of the final volume) were then used in One-step RT-qPCR reactions (Bioline) utilizing the E4F (forward) and E4R (reverse) primer pair (sequences shown in S2 Table in S1 File). An additional RT-qPCR reaction containing 10 fmoles of the template DNA was included to normalize Ct values. Results were normalized to the E4 transcript levels from the reactions without any recombinant AR or ARv7 protein added (i.e., basal transcription) to determine relative E4 mRNA fold change.

## Immunoblot analysis

Immunoblotting was performed essentially as we have previously described [40]. Primary antibody diluted in blocking buffer (5% nonfat milk and PBS) was incubated with PVDF blot strips for at minimum 2 hours to overnight at 4°C, followed by at least three PBS washes. Secondary HRP-conjugated antibodies in blocking buffer were added (sheep anti-mouse or donkey anti-rabbit, GE Healthcare; rabbit anti-goat, Santa Cruz) for 1 hour followed by at least three final PBS washes. Signal was developed on X-ray film via ECL Plus. Cropped immunoblot images were quantified via Image J (free download from NIH) where cited. All raw, uncropped immunoblot images are provided in S6 Fig.

## Mass spectrometry analysis

ARE pulldown samples in SDS sample buffer were submitted to the BCM Mass Spectrometry Proteomics Core to be analyzed by liquid chromatography-mass spectrometry (LC-MS) as described previously [40].

## siRNA knockdowns

LNCaP cells or LNCaP dox-inducible ARv7 cells were plated in 12-well dishes (for gene expression analyses) at $1 \times 10^6$ cells per well in 10% fetal calf serum containing RPMI 1640 medium as indicated above and then medium was exchanged with charcoal stripped 10% fetal calf serum containing RPMI 1640 medium. Cells were then transfected in triplicate with 50 nM of targeting siRNA or a non-targeting siRNA using TransIT-TKO transfection reagent for 2 days. For gene expression analyses, where required, R1881 or dox was added to cells after two days of knockdown and were harvested 24

hours later. siRNAs for ANP32A and CAND1 were custom designed in sets of three by Sigma-Aldrich before being pooled together. The non-targeting control used with ANP32A and CAND1 was Mission siRNA Universal negative control #1 (Sigma-Aldrich). These siRNAs were validated for knockdown at the mRNA level by RT-qPCR. siRNAs targeting DNA-PKcs were from Santa Cruz Biotechnology and previously validated [40], with siRNA-A as the non-targeting control. siR-NAs for SRC-3 were custom designed in sets of three by Sigma-Aldrich before being pooled together and were previously validated [46]. All siRNA sequences are detailed in S3 Table in S1 File.

### DNA-PK inhibition

LNCaP cells or LNCaP dox-inducible ARv7 cells were plated in 12-well dishes as above. After two days, cells were treated with the DNA-PK inhibitor NU7441 for 1 hour before adding R1881 or dox. After 24 hours the cells were harvested for RNA purification and cDNA synthesis. NU7441 was dissolved in DMSO and stored at −20°C.

### Gene expression analysis

RNA isolated from a well of a 12-well plate of LNCaP cells was harvested using Tri-reagent, following the manufacturer's instructions. The RNA concentrations were quantified by Nanodrop (ThermoFisher Nanodrop Lite). One µg of each RNA sample was used to make cDNAs by First-Strand cDNA Synthesis using SuperScript II Reverse Transcriptase. cDNAs (20 µl) were then diluted with 180 µl of DEPC-treated water. To analyze gene expression, 2 µl of cDNAs were added to qPCR reactions along with Taqman Universal master mix II (Applied Biosystems), 200 nM primers (designed using Roche Diagnostics Universal ProbeLibrary System Assay Design), and the appropriate UniversalLibrary Probes, on a StepOnePlus machine (Applied Biosystems). Gene expression was normalized to human *GAPDH* mRNA and relative mRNA level was determined by the delta $C_t$ method [47]. Primers used for qPCR are shown in S4 Table in S1 File.

### Chromatin immunoprecipitation (ChIP) assays

LNCaP dox-inducible ARv7 cells were grown in 15 cm dishes as above and treated (-/+) dox or (-/+) R1881 as indicated in Fig 4 legend. Cells were then crosslinked with 1% formaldehyde at room temperature, quenched with 125 mM glycine, and then harvested by scraping in cold PBS (containing Xpert protease inhibitors (GenDEPOT)). Lysates were prepared in 250 µl FA lysis buffer [42] with protease inhibitors, frozen in dry ice/ethanol bath for 5 min, thawed, and then chromatin was sheared in a water bath sonicator (Bioruptor from Diagenode). Samples were sheared at 30% amplitude for 10 cycles of 20 seconds on and 20 seconds off and then centrifuged for 5 min at 13,000 rpm. 60 µl of supernatant was aliquoted for quality control check for sizes of DNA fragmentation as revealed on agarose gels. 180 µl of supernatant was diluted 10-fold with ChIP dilution buffer (0.55% Triton X-100, 1.2 mM EDTA pH 8.0, 16.8 mM Tris pH 8.0, 167 mM NaCl) containing protease inhibitors. Input controls were moved to separate tubes (150 µl of the lysate) and the remainder was pre-cleared with 25 µL of Protein A/G Magnetic Dynabeads conditioned with 0.5 µg/µL sheared salmon sperm DNA for 30 min at 4°C with rotation. Immunoprecipitations were performed with 50 µL of the above Dynabeads using 2 µg of AR, CAND1, or control rabbit IgG antibodies incubated overnight at 4°C with rotation. After serial washes (each done two times and sequentially) in 1 mL ice-cold low salt TSE I buffer [37], high salt TSE II buffer [37], and TE buffer (10 mM Tris-Cl pH 8.0 and 1 mM EDTA pH 8.0), the precipitated DNA was eluted three times from the beads by incubation in 50 µL elution buffer (1% SDS, 100 mM NaHCO$_3$) at room temperature for 10 min. Crosslinking of pooled eluted DNA was reversed by incubating eluants for at least 6 hours at 65°C and DNA fragments were purified on spin columns from a PCR purification kit. Two µL out of 90 µL eluted ChIP samples were quantified by qPCR using SYBR green chemistry and primers flanking known AREs of the *PSA* gene enhancer or the *FKBP5* promoter. Enrichment of AR/ARv7 or CAND1 at AREs of *PSA* gene enhancer or *FKBP5* gene was normalized to input chromatin and presented as percentage of input chromatin as we have described [42]. Relative quantification of each sample was performed against a standard curve plotted using serial dilution of pooled input. S2 Table in S1 File details ChIP-qPCR primer sequences.

### Bioinformatics analysis

The expression of *CAND1* mRNA across normal, clinically localized primary prostate tumor and metastatic tumor samples was evaluated using two prostate cancer patient cohorts: Varambally and colleagues ([48], GSE3325) and Cai and colleagues ([49], GSE32269). Further, clinical relevance of *CAND1* mRNA expression was evaluated with The Cancer Genome Atlas (TCGA-PRAD) and Glinsky [50] prostate cancer cohorts. Patients in both cohorts were divided into top 45% and bottom 45% according to the *CAND1* mRNA expression and significance was assessed via survival package in R statistical system with log-rank test (P<0.05) and the Cox proportional hazard test (P<0.05).

### Immunodepletions from nuclear extract

Our detailed protocol for immunodepletion of antigens from HeLa NE was published [40]. Immunodepletion of candidates was confirmed by immunoblotting. To immunodeplete DNA-PKcs, 16 μg of mouse DNA-PKcs antibody or 16 μg mouse IgG were incubated with 2 mg of HeLa NE (S1 Table in S1 File provides antibody details) and three rounds of immunodepletion were done.

### Statistical analysis

Representative experiments are shown in this manuscipt, but each was done three independent times. Within an experiment, single comparisons were carried out using a two-tailed, unpaired Student t-test in Excel. Error bars in each figure represent the standard error of the mean (SEM) from independent biological triplicates, unless noted in a particular figure legend. *p*<0.05 was considered statistically significant. Statistical significance for the bioinformatics data was tested as described above.

## Results

### Establishing an ARE pulldown assay to characterize ARv7 CoR complexes

The human *AR* gene is encoded by 8 exons: exon 1 encodes the N-terminal domain (NTD) that contains the potent AF1 transactivation domain [51–53], exons 2 and 3 encode the DNA binding domain (DBD), exon 4 encodes the hinge region, which includes the nuclear localization signal, and exons 4–8 encode the LBD. ARv7 is truncated after exon 3 and contains 16 unique amino acids from cryptic exon 3 (Fig 1A, *top*). To compare the transcriptional activity of ARv7 with AR on a defined promoter driven by three AREs, cell-free transcription assays were developed and employed. An 853 bp DNA construct containing three AREs from the rat *TAT* gene coupled to the *Adenovirus* E4 minimal promoter (3xARE-E4) was assembled into chromatin (via salt dilution) using purified HeLa core histones. The chromatinized DNA template was then incubated with human recombinant purified AR or ARv7 protein, HeLa S3 cell nuclear extract (NE) as a source of the RNA polymerase II (pol II) transcriptional machinery and relevant CoRs, and the potent synthetic agonist R1881 (for AR activation) or DMSO vehicle (for ARv7). After allowing the pre-initiation complex to form, ribonucleotide triphosphates (NTPs) were added to initiate transcription. The transcriptional activity of AR and ARv7 was determined via RT-qPCR by directly measuring the output of synthesized E4 transcripts. As expected, an increasing amount of AR or ARv7 led to enhanced E4 mRNA transcripts; however, significantly less ARv7 protein was required to produce the same transcriptional output as AR (Fig 1A, *bottom*). Importantly, these data are consistent with prior cell-based assays measuring expression of an ARE-dependent reporter gene as a function of expressed AR or ARv7 [12].

To identify proteins that interact with ARv7 and AR, an ARE pulldown assay was employed that utilized a biotinylated 3xARE-E4 fragment produced by PCR from the template used above for cell-free transcription (Fig 1B). The DNA was bound to magnetic streptavidin coated beads, then pre-incubated with recombinant ARv7 or AR protein to pre-bind it to the AREs to limit other TFs binding the AREs. We first tested how the recombinant ARv7 would bind DNA versus AR by titrating various amounts with a set amount of 3xARE DNA-beads. At amounts tested, similar ARv7 and AR DNA binding

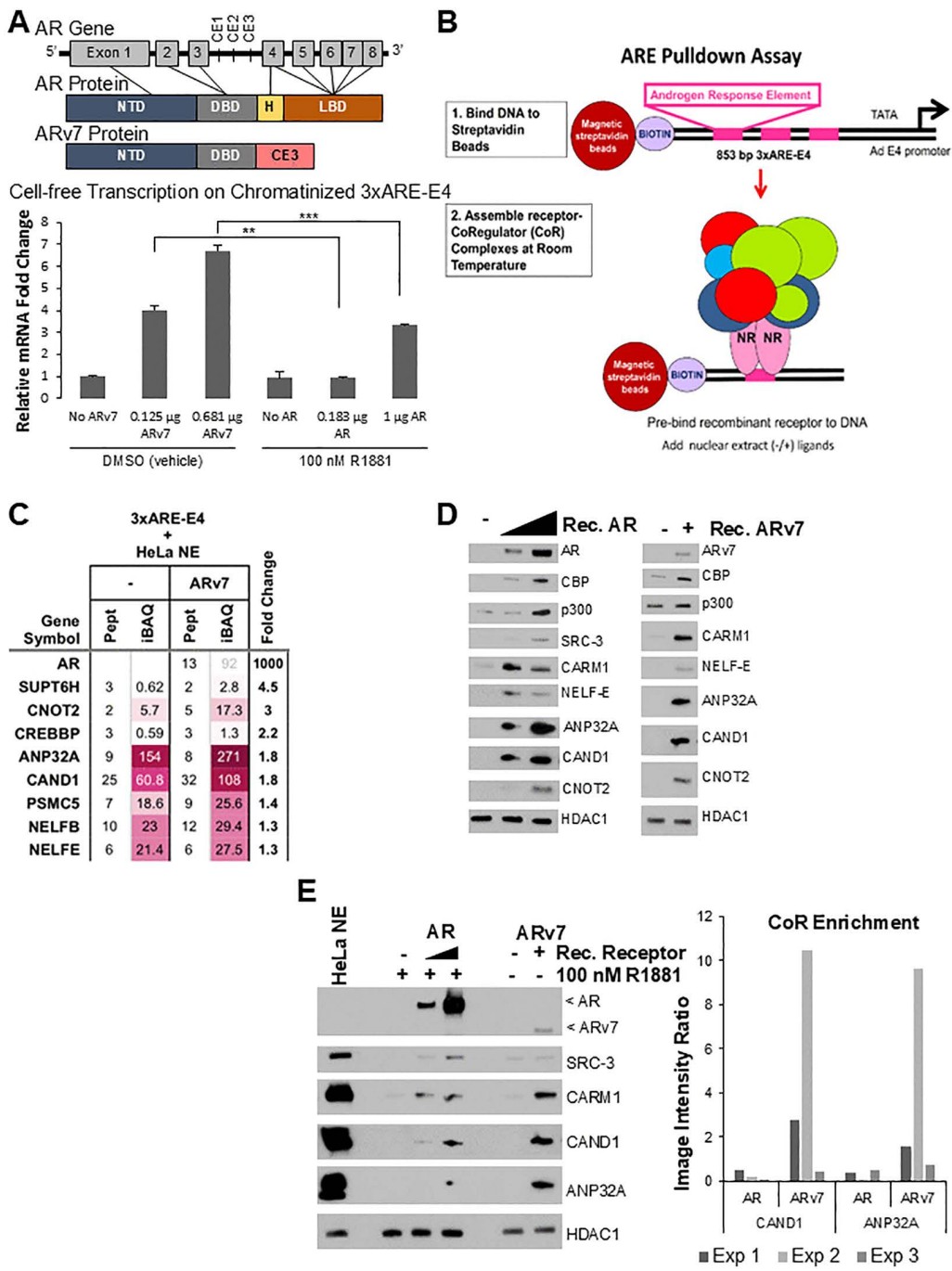

**Fig 1. Proteomic analyses from ARE pulldowns identify ANP32A and CAND1 as previously undescribed ARv7 coregulators. (A)** *Top*, Schematic demonstrating the human *AR* gene and resultant AR and ARv7 protein domain structures. CE1-3 denotes cryptic exons 1-3. *Bottom*, Cell-free transcription demonstrates ARv7 has substantially higher cell-free transcriptional activity at a 1:1 molar ratio with AR. **$p = 0.0067$, ***$p = 0.00021$. **(B)** Schematic of ARE pulldown showing biotinylated 3xARE-E4 immobilized on streptavidin coated beads and incubated with purified AR or ARv7, NE, and ligands to form the receptor-CoR complexes on DNA. **(C)** Heatmap of the MS data for (-/+) ARv7 ARE pulldowns illustrating potential "transcriptional coregulators" recruited with recombinant ARv7 that were subsequently validated by immunoblotting. CBP (Gene symbol: CREBBP) is a known AR coactivator [54,55], while the other proteins shown represent potential new ARv7 coregulators. "Fold change" represents the amount (iBAQ) of a CoR with ARv7 divided by the amount of that CoR with no receptor added. **(D)** Immunoblotting validation of MS candidates in both AR and ARv7 DNA pulldowns using both HeLa NE plus recombinant AR (1 μg, 2 μg) or ARv7 protein (0.25 μg). Known AR coactivators [29] recruited to AR in this assay include CBP, p300, SRC-3,

and CARM1. HDAC1 was used as a loading control, as its level on DNA beads was not receptor- or ligand-dependent. **(E)** *Left*, Representative AR (0.5 µg, 2 µg) and ARv7 (0.25 µg) ARE pulldowns run on the same SDS-PAGE gel demonstrate enhanced recruitment of a subset of AR-associating proteins with ARv7 such as CARM1, CAND1, and ANP32A. HDAC1 was used as a loading control. All HeLa NE ARE pulldowns with recombinant AR had 100 nM R1881 present. Note that the N-terminal AR antibody 441 was employed such that a true comparison of AR to ARv7 protein bound levels could be determined. *Right*, Quantification of three independent AR versus ARv7 ARE pulldowns demonstrating increased ANP32A and CAND1 enrichment with ARv7 compared to AR. Image J was used for quantification of the ratio of ANP32A or CAND1 band signal normalized to AR or ARv7 bound to the AREs (image intensity ratio). Exp: Experiment.

were observed (S1A Fig in S1 File). The beads were then incubated with HeLa S3 cell NE as a source of CoRs, and agonist ligand (R1881) for AR. After stringent washes to minimize non-specific background protein binding to the DNA, the samples were assayed for ARv7 and AR remaining on the beads. We found that much less ARv7 and AR were bound after incubation with NE and washes (S1A Fig in S1 File, see asterisks). NE from the HeLa S3 cell line is the "workhorse" used because it is an excellent source of RNA pol II machinery and CoRs and it also does not express AR or ARv7, allowing for controlled ARE pulldowns (-/+) purified recombinant protein to determine if CoR recruitment is indeed AR- or ARv7-dependent. We next tested which amount of ARv7 or AR to add into the system for optimal recruitment of a known AR coactivator, CARM1 [56]. From titration assays, 2.0 µg AR and 0.25 µg ARv7 revealed the best recruitment of CARM1 to the ARE-containing beads (S1B Fig in S1 File, *left* panel). Interestingly, unlike AR, we found that increasing ARv7 to 1 µg in the ARE pulldowns led to a reduction in both CARM1 and another known AR coactivator, E6-AP [57] (S1B Fig in S1 File, *right* panel). Thus, we determined from these titration experiments the optimal amounts of ARv7 (0.25 µg) and AR (2 µg) to add to the same amount of 3xARE-beads and HeLa NE for recruitment of coregulators.

As HeLa cells are derived from a cervical carcinoma, some coregulators recruited to AR or ARv7 could be specific to this line and not a prostate cancer cell. To begin to address this concern, we performed ARE pulldowns from nuclear extracts of LNCaP prostate cancer cells that endogenously express AR. We found that endogenous AR when incubated with R1881 recruited known coactivators (CBP, SRC-3, and CARM1) from prostate cancer cells (S1C Fig in S1 File) similar to when HeLa NE and recombinant AR is employed (see below).

### Mass spectrometry identified ANP32A and CAND1 as previously undescribed ARv7 associated proteins that display enhanced recruitment with ARv7 over AR

As there are many known AR interacting CoRs already (e.g., [29]) and significantly less is known about ARv7, ARE pulldowns were submitted for unbiased screening analyses by liquid chromatography (LC)-MS analysis (-/+) ARv7 protein in order to identify and characterize the ARv7 recruited complexes. The MS results identified many proteins that were recruited in an ARv7-dependent fashion, including the known AR coactivator CBP (gene symbol: CREBBP) [54,55] (Fig 1C; S5 Table provides the full searchable list of all bound proteins). In addition to CBP, Fig 1C shows seven additional MS candidates characterized as potential "transcriptional coregulators" based on our prior extensive analysis [58] that were recruited by ARv7 and further validated by immunoblotting (Fig 1D; S1D Fig in S1 File). Some of the candidates have not been previously described to play a role in AR signaling (e.g., CCR4-NOT complex subunit CNOT2, RNA pol II negative elongation factor complex subunit NELF-E, RNA pol II elongation factor SUPT6H (also called SPT6), 19S proteasome subunit PSMC5 (also called SUG1), and cullin-associated NEDD8-dissociated protein 1 (CAND1, also called TIP120 [59]) or with limited data as an AR coactivator, acidic leucine-rich nuclear phosphoprotein 32 family member A (ANP32A) [60] or as an AR corepressor, NELF-B (also known as cofactor of BRCA1 or COBRA1 [61]). Many of these potential CoRs also were recruited by R1881-bound AR from HeLa NE (Fig 1D) and LNCaP NE (S1C Fig in S1 File).

When directly comparing AR- and ARv7- CoR complexes by immunoblotting, ARv7 demonstrated a noticeably enhanced recruitment of a subset of AR-associated proteins including ANP32A, CAND1, and CARM1 (Fig 1E, with

quantification of CAND1 and ANP32A recruitment relative to AR isoform levels in three experiments shown in the *right* panel). As CARM1 is a known AR coactivator, we chose to focus on determining whether ANP32A and CAND1 are, indeed, AR/ARv7 coregulators. Notably, significantly less ARv7 protein was required to recruit these specific candidate coregulators compared to AR, similar to the above cell-free transcription data (Fig 1A).

**Knockdown of ANP32A or CAND1 in LNCaP and LNCaP doxycycline-inducible ARv7 cells demonstrates coactivator roles for AR and ARv7 target gene expression**

To determine the functional relevance of ANP32A and CAND1 for AR transcriptional activity, LNCaP cells were used to investigate effects on AR target gene expression. LNCaP cells were grown in charcoal stripped serum (CSS) to prevent activation of the endogenous AR by the endogenous androgens in FBS and to establish a baseline for AR target gene expression. The cells were then treated with small interfering RNAs (siRNAs) targeting known AR coactivators (MED1 [62] or SRC-3 [23]) or our candidates (ANP32A or CAND1) for 48 hours for efficient knockdown followed by the addition of 10 nM R1881 for 24 hours. One siRNA non-targeting sample (Ctrl) was also treated with the AR antagonist enzalutamide (formerly called MDV3100) for 1 hour prior to the addition of R1881 as a quality control to demonstrate inhibition of AR target gene expression. The inclusion of SRC-3 and MED1 knockdown serve as positive controls as their knockdown in LNCaP cells has been reported to reduce the expression of the AR target gene, *PSA* [38,63]. As expected, SRC-3 and MED1 knockdown significantly reduced R1881-induced *PSA* expression. Interestingly, knockdown of ANP32A or CAND1 also significantly reduced the R1881-induced expression of *PSA* (Fig 2A). These results suggest ANP32A and CAND1 may function as coactivators of androgen-liganded AR transcriptional activity.

In order to test the functional relevance of ANP32A and CAND1 with ARv7, doxycycline (dox)-inducible LNCaP cells [42] were employed. In the presence of dox, the tet repression of ARv7 is released allowing for *ARv7* mRNA expression and ARv7 induction of its specific target genes (such as *EDN2* that is not induced by AR [42]) (S2A Fig in S1 File). When the LNCaP-ARv7 cells are grown in CSS serum, they can be treated with: 1) R1881 to investigate endogenous AR-driven target gene expression or 2) dox to investigate ARv7-driven target gene expression with minimal impact from the endogenous AR (Fig 2B). Robust siRNA mediated knockdown of ANP32A and CAND1 significantly reduced the AR (+R1881) driven expression of *PSA* and *EXTL2* (Fig 2C; S2B Fig in S1 File) with no effect on basal expression. *PSA* is a well-known target gene of AR (and ARv7), and *EXTL2* was previously reported to be an AR-specific target gene [42]. Additionally, the knockdown of ANP32A and CAND1 significantly reduced the ARv7-driven expression of its specific target genes *EDN2* and *ETS2* [42] (Fig 2D). As a control, *ARv7* mRNA expression was measured upon ANP32A or CAND1 knockdown and it was not reduced in either case, suggesting the reduced ARv7 target gene expression was not simply due to a loss of *ARv7* mRNA expression (S2C Fig in S1 File). Lastly, we tested the effect of dual knockdown of ANP32A and CAND1 on AR and ARv7 target gene expression. The combination knockdown significantly further reduced the gene expression of AR and ARv7 target genes (S3 Fig in S1 File). Collectively these results suggest that the newly identified AR and ARv7-associating proteins, ANP32A and CAND1, function as AR and ARv7 coactivators.

**Bioinformatic analyses of CAND1 expression in human prostate cancer clinical datasets demonstrate significant correlation with prostate cancer progression, metastasis, and worse patient prognosis**

Results from the above cell-free and cell-based assays suggest ANP32A and CAND1 have coactivator functions for AR and ARv7 and may play a role in prostate cancer. To address this, the expression of *ANP32A* and *CAND1* mRNA was evaluated in two human prostate cancer clinical datasets made up of clinically localized primary and metastatic tumor samples (Varambally and colleagues [48]; Cai and colleagues [49]). *CAND1* mRNA expression significantly correlated with metastatic prostate cancer progression in both datasets (Fig 3A, *left* and *middle*). ANP32A did show an increase in mRNA expression between normal and primary prostate cancer when evaluated with The Cancer Genome Atlas (TCGA)-PRAD; however, this increase was not statistically significant (Fig 3A, *right*). Furthermore, the clinical relevance of *CAND1*

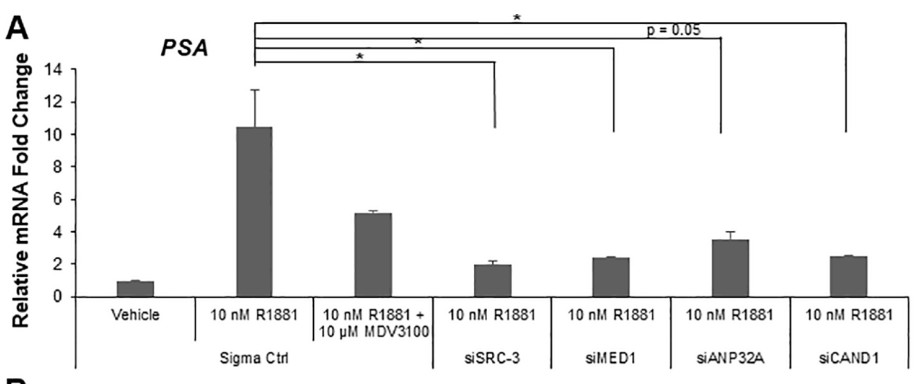

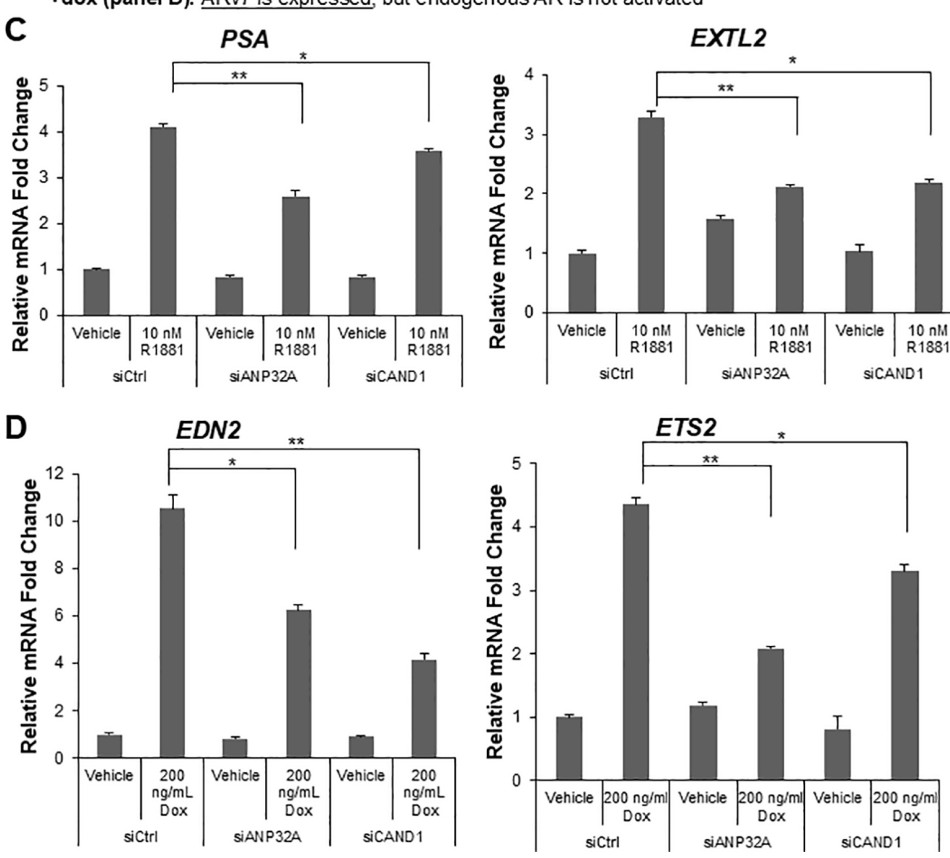

**Fig 2. Knockdown of CAND1 and ANP32A reduces AR and ARv7 target gene expression in parental LNCaP and LNCaP doxycycline (dox)-inducible ARv7 cells. (A)** LNCaP cells grown in charcoal stripped serum (CSS) were transfected with siRNAs targeting SRC-3 and MED1 (known AR coactivators) and ANP32A and CAND1 (candidates identified from MS and immunoblotting) for 48 hours. For enzalutamide (previously called MDV3100) treatment the cells were treated with 10 μM MDV3100 for 1 hour before adding 10 nM R1881. Knockdown of all four proteins significantly reduced R1881-dependent expression of the *PSA* gene. P-values in this panel: for siSRC-3 = 0.028, for siMED1 = 0.032, for siANP32A = 0.050, and for siCAND1 = 0.034. **(B)** Schema for experiments with LNCaP dox-inducible ARv7 cells. **(C)** Knockdown of ANP32A and CAND1 in LNCaP dox-inducible ARv7 cells (only treated with R1881 and no dox) significantly reduced AR-driven target gene expression of *PSA* and *EXTL2*. P-values for *PSA* data: siANP32A = 0.0020; siCAND1 = 0.037. P-values for *EXTL2* data: siANP32A = 0.0049; siCAND1 = 0.016. **(D)** ANP32A and CAND1 knockdown significantly reduced expression of two dox-induced ARv7-driven target genes. *EDN2* and *ETS2* are published targets of ARv7 [42]. In panels C and D, cells were grown in CSS serum and treated with siRNAs for 48 hours before adding 10 nM R1881 or dox (200 ng/ml).

P-values for *EDN2* data: siANP32A = 0.020; Ctrl to siCAND1: 0.0098. P-values for *ETS2* data: siANP32A = 0.0010; siCAND1 = 0.049. In all panels, gene expression was analyzed by RT-qPCR and normalized to human *GAPDH* mRNA levels. *$p < 0.05$, **$p < 0.01$.

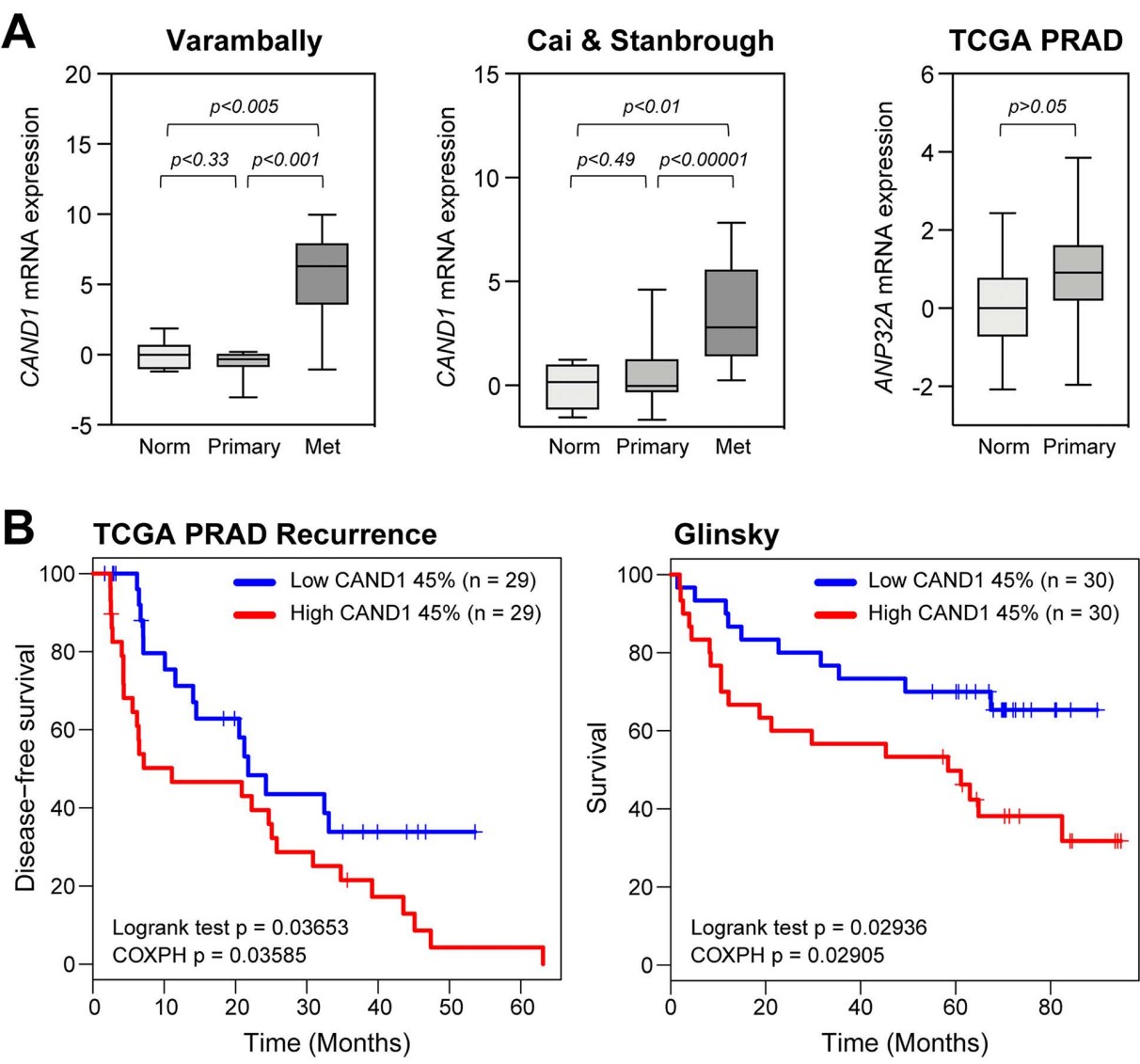

**Fig 3. *CAND1* gene expression correlates with prostate cancer progression and worse patient prognosis. (A)** *Left* and *middle*, *CAND1* mRNA expression was evaluated in two human prostate cancer clinical datasets (Varambally *et al.* [48] and Cai *et al.* [49]). For each boxplot, the line shows the median, with box identifying interquartile range and whiskers showing min/max values. Higher expression of *CAND1* correlated with metastatic prostate cancer. *Right*, *ANP32A* mRNA expression was evaluated in the TCGA dataset, showing no significant difference between normal and primary prostate cancers ($p > 0.05$). Norm indicates normal prostate, Primary indicates primary tumor, and Met indicates metastatic prostate cancer. Pairwise t-tests for each comparison determined significance, as indicated by the p-values. **(B)** Bioinformatics analyses using TCGA and Glinsky *et al.* [50] human prostate cancer clinical datasets evaluated *CAND1* mRNA expression with disease-free survival and prostate cancer patient survival, respectively. Blue indicates low *CAND1* mRNA expression and red indicates high *CAND1* mRNA expression. Significance was assessed via survival package in R statistical program with log-rank test ($p < 0.05$) and the Cox proportional hazard test ($p < 0.05$).

gene expression was evaluated with The Cancer Genome Atlas (TCGA)-PRAD and Glinsky [50] prostate cancer cohorts. Patients in both cohorts were divided into top 45% and bottom 45% according to the *CAND1* gene expression. These analyses demonstrated with significance that high *CAND1* mRNA expression correlated with increased risk of recurrence and poorer patient survival (Fig 3B). Collectively, these findings suggest that CAND1 may contribute to driving prostate cancer progression and metastasis through modulating AR and ARv7 activity.

### CAND1 co-localization with AR and ARv7 at target genes in LNCaP doxycycline-inducible ARv7 cells

To further validate the functional interaction of CAND1 with AR and ARv7 at endogenous gene AREs, we conducted chromatin immunoprecipitation (ChIP) studies in LNCaP dox-inducible ARv7 cells using AR or CAND1 antibodies (Fig 4A). As expected, using an N-terminal antibody for AR that recognizes both AR and ARv7, ChIP in these cells demonstrated R1881-induced AR localization at the *PSA* enhancer (PSAenh) and *FKBP5* gene promoter AREs. When the cells were treated with dox to induce ARv7 expression, ChIP assays demonstrated that ARv7 localized at these same loci, albeit with less occupancy as compared to AR (Fig 4A). Importantly, CAND1 ChIP assays revealed that CAND1 co-localized at the same AREs with AR and ARv7 either in response to R1881 or dox, respectively (Fig 4A). Furthermore, because the same N-terminal antibody was used for AR and ARv7 ChIP, direct comparisons can be made between them. Taking the ratio of CAND1 enrichment over AR or ARv7 enrichment suggests that smaller amounts of chromatin-bound ARv7 can recruit more CAND1 than chromatin-bound AR; this is especially evident at the *FKBP5* gene ARE (Fig 4B). Collectively, these results indicate that CAND1 functions as a coactivator of both AR proteins and that ARv7 has a greater capability for recruiting this coactivator to these gene loci than AR.

### DNA-PK dynamically regulates the AR- and ARv7-CoR complexes and stabilizes their interactions with ANP32A and CAND1

Although the ARE pulldown assay primarily is used to identify the "core or stable" CoR complexes recruited with AR and ARv7 to DNA, it also can be used for determining if regulatory enzymes, such as kinases, may modify the ARs themselves or alter their CoR complexes. To determine if there were any kinases that regulated the AR complex, the ATP treatment (or kinase) assay was employed. Here, the standard ARE pulldown is conducted (-/+) recombinant AR protein. After final washes, the DNA-bound AR-CoR complexes are pelleted by magnet and resuspended in 1x kinase buffer. Upon the addition of ATP, the reactions are moved to 30°C for up to 30 min. Interestingly, gel mobility shifts of AR were detected by SDS-PAGE within 5 min of ATP addition (Fig 5A).

Typically, protein gel mobility shifts indicate a post-translational modification has occurred (e.g., phosphorylation). To determine if the AR gel shifts might indeed be due to phosphorylation, the AR-CoR complexes were treated with ANP-PMP (a non-hydrolyzable form of ATP) to compare with ATP, or ATP-treated complexes were then treated with λ protein phosphatase (LPP) to counteract any phosphorylation events due to kinases. Strikingly, the AR gel mobility shifts were significantly attenuated in both the ANP-PMP and LPP-treated reactions (Fig 5B), indicating that AR was being phosphorylated by a kinase. Considering that DNA-dependent protein kinase (DNA-PK) has been shown to be involved with AR signaling and prostate cancer progression [41,64], another ATP treatment reaction was included with NU7441, a highly specific and potent DNA-PK inhibitor [65]. Inhibiting DNA-PK with NU7441 significantly reduced the AR gel mobility shifts indicating that DNA-PK was at least one of the kinases targeting AR (Fig 5B).

To further support this finding, the catalytic subunit of DNA-PK (DNA-PKcs) was immunodepleted from HeLa NE to investigate the effects of losing DNA-PK on AR phosphorylation. Loss of DNA-PK significantly increased the amount of unphosphorylated AR when compared to the IgG control NE reactions (Fig 5C). Because there are many kinases from the HeLa NE that could be phosphorylating AR, the ATP treatment assay was repeated using purified proteins only. Incubating 3xARE-E4 DNA with recombinant AR, recombinant DNA-PK holoenzyme (DNA-PKcs in complex with Ku70/Ku80) with or without ATP demonstrated an ATP-dependent AR gel mobility shift, and this shift was significantly reduced in the presence

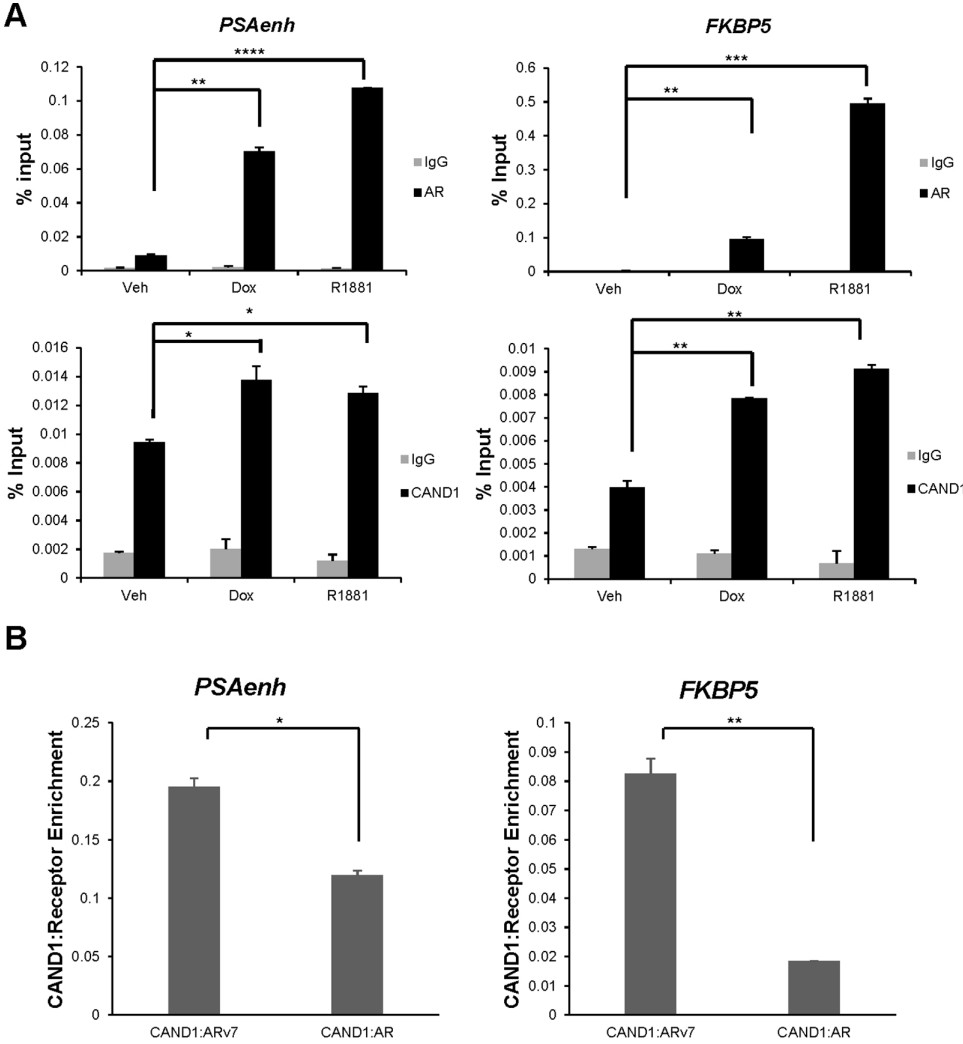

**Fig 4. AR/ARv7 and CAND1 protein co-localize to AREs of two AR/ARv7 common induced genes in LNCaP dox-inducible ARv7 cells. (A)** ChIP assays were performed in LNCaP dox-inducible ARv7 cells grown in CSS followed by dox (20 ng/mL) or R1881 (10 nM) treatment with either IgG control, AR antibody, or CAND1 antibody. ChIP for AR and ARv7 utilized an N-terminal antibody that detects both receptors so direct comparisons could be made between their enrichment at either gene loci. Enrichment of AR/ARv7 or CAND1 at AREs of *PSA* gene enhancer or *FKBP5* gene was normalized to input chromatin and presented as percentage of input chromatin. P-values: for PSAenh *top*: veh to dox, p = 0.0010; veh to R1881, p = 0.000080; PSAenh *bottom*: veh to dox, p = 0.046; veh to R1881, p = 0.016; for FKBP5 *top*: veh to dox, p = 0.0035; veh to R1881, p = 0.00070; for FKBP5 *bottom*: veh to dox, p = 0.0050; veh to R1881, p = 0.0040. **(B)** The ratios of mean CAND1 ChIP enrichment to either ARv7 or AR mean ChIP enrichment are presented for the *PSA* gene enhancer and *FKBP5* gene promoter. P-values: for PSA enh, p = 0.012; for FKBP5, p = 0.0063. *p < 0.05, **p < 0.01, ***p < 0.001, ****p < 0.0005.

of NU7441 (Fig 5D, *left* panel). Additionally, ARv7 gel mobility shifts were seen when purified ARv7 was tested in place of AR (Fig 5D, *right* panel). Collectively, these data suggest that DNA-PK directly phosphorylates both androgen receptors.

After establishing AR/ARv7 as a DNA-PK target, the relationship between DNA-PK and other components of the AR- and ARv7-CoR complexes was investigated. Interestingly, upon ATP-treatment the interaction of CAND1 and ANP32A in the AR-CoR complexes was noticeably stabilized during the 30°C treatment period and this stabilization was reduced in the presence of NU7441 (Fig 5E, *left* panel). This suggests that DNA-PK enzymatic activity is important for stabilizing the association of ANP32A and CAND1 within the AR complex. Furthermore, ATP treatment of

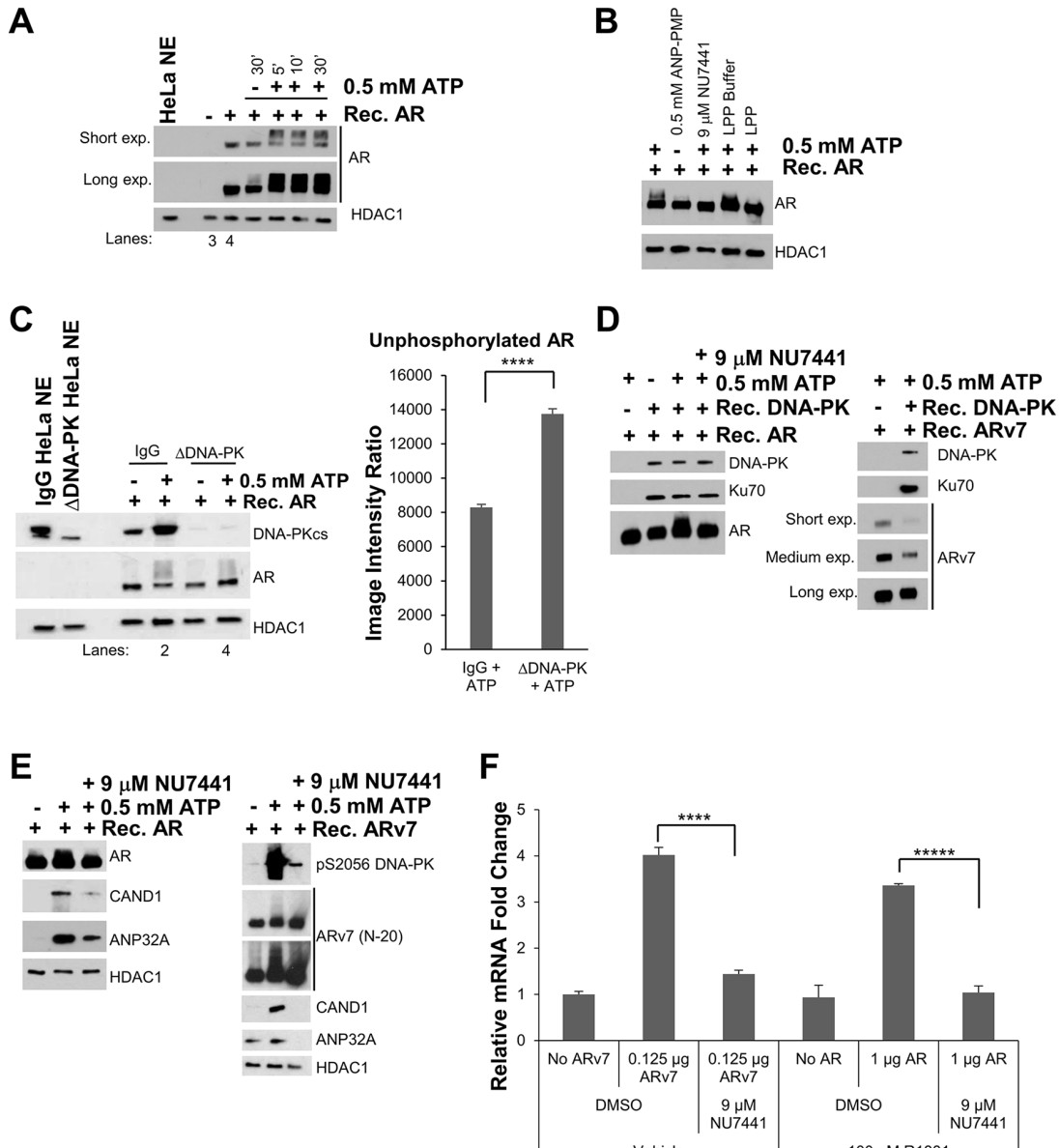

**Fig 5. DNA-PK action stabilizes the interaction of CAND1 and ANP32A in AR- and ARv7-coregulator complexes and DNA-PK functions as an AR and ARv7 coactivator in cell-free transcription assays. (A)** After standard ARE pulldowns, the AR-DNA-bound complexes were washed and resuspended in 1x kinase buffer with/without ATP at 30°C for listed times. Gel mobility shifts in AR were detected as a function of ATP addition. All pulldowns had 100 nM R1881 present. Proteins were analyzed by immunoblotting. Lanes 3 and 4 represent the standard AR-ARE pulldown. **(B)** To test if the above AR gel mobility shifts were reflective of phosphorylation events, the AR-CoR complexes were treated with ATP or AMP-PNP (non-hydrolyzable analogue) for 30 min at 30°C. The DNA-PK inhibitor NU7441 or λ protein phosphatase (LPP) were pre-incubated for 5 min before adding ATP. LPP buffer was used for the LPP control reaction. **(C)** DNA-PK plays a role in phosphorylating AR. ATP treatment of the AR-CoR complexes using HeLa NE where DNA-PKcs was immunodepleted reveals a reduction of AR gel mobility shifts (*left* panel) with the loss of DNA-PK. Proteins were assayed by immunoblotting and the level of unphosphorylated AR was quantified by Image J after normalization to the loading control HDAC1 (*right* panel). P-value here = 0.00011. **(D)** DNA-bound AR or ARv7 was incubated with purified heterotrimeric DNA-PK (i.e., DNA-PKcs and Ku70/80) and then treated (-/+) ATP (-/+) NU7441 to demonstrate direct phosphorylation of AR/ARv7 by DNA-PK. Gel mobility shifts were assayed by immunoblotting. **(E)** ATP treatment (-/+) NU7441 pre-treatment demonstrates that DNA-PK enzymatic activity is needed for stabilizing CAND1 and ANP32A in the AR/ARv7 complexes. Auto-phosphorylation of DNA-PKcs at serine 2056 (pS2056) is a marker of DNA-PK activity and its inhibition by NU7441. Proteins were assayed by immunoblotting. ARv7 was detected here with the N-20 antibody. **(F)** HeLa NE used in 3xARE-E4 cell-free transcription assays was pre-treated (-/+) NU7441 to determine effects on AR and ARv7 transcriptional activity. E4 mRNA transcript levels were measured by RT-qPCR. In panels A-C

and E, HDAC1 was used as a loading control and ARE pulldowns were conducted using HeLa NE as listed. All AR reactions had 100 nM R1881 present. Short exp., Medium exp., and Long exp. denote short, medium, and long exposures during x-ray film development. P-values: for ARv7, p = 0.00015; for AR, p = 0.000035. ****$p < 0.0005$, *****$p < 0.0001$.

ARv7-CoR complexes demonstrated a similar effect with ANP32A and CAND1 being stabilized in the complex, and upon NU7441 treatment, this effect was similarly lost (Fig 5E, *right* panel). Additionally, as a quality control, a phospho-serine DNA-PKcs antibody was used to immunoblot serine 2056 (pS2056) of DNA-PK. DNA-PK can auto-phosphorylate itself at this residue and it is known to be an activating mark [66]. ATP treatment increased phosphorylation of serine 2056 and inhibiting DNA-PK with NU7441 reduced it, as expected (Fig 5E, *right* panel). Collectively, these findings suggest that DNA-PK stabilizes the interaction of ANP32A and CAND1 in the biochemically isolated AR- and ARv7-CoR complexes.

### DNA-PK functions as a coactivator of AR and ARv7 target genes both in cell-free transcription and cell-based assays

Given the above data of DNA-PK phosphorylating AR/ARv7 and stablizing ANP32A and CAND1 in the AR/ARv7 complexes, we next asked what effect the loss of DNA-PK activity would have on AR/ARv7 driven cell-free transcription. We observed reductions in both AR and ARv7-driven transcription when NU7441 was added to the reactions (Fig 5F), directly showing that DNA-PK stimulates AR and ARv7 transcription *in vitro*. Importantly, DNA-PK inhibition with NU7441 did not have any repressive effect on basal transcription, thereby ruling out any trivial effects on the RNA pol II transcriptional machinery (S4A Fig in S1 File).

To further support these cell-free transcription findings, the role of DNA-PK was tested in LNCaP dox-inducible ARv7 cells grown in CSS serum. The cells were transiently transfected with DNA-PKcs siRNAs for 48 hours before being treated with R1881 or dox to test effects of DNA-PK depletion on both AR and ARv7 target gene expression. Knockdown of DNA-PKcs (controls shown in S4B Fig in S1 File) significantly reduced the expression of *PSA* and *FKBP5*, which are shared AR and ARv7 target genes (Fig 6A, *left* and *middle* panels). Loss of DNA-PKcs also significantly reduced *EXTL2* and *RASSF3*, two AR-specific target genes [42] (S5A Fig in S1 File). *EDN2*, an ARv7-specific target gene [42], also displayed significant reduction in expression upon DNA-PKcs knockdown (Fig 6A, *right* panel). Importantly, DNA-PK depletion did not reduce, but actually increased, *ARv7* mRNA levels (S5B Fig in S1 File). In sum, loss of DNA-PK diminishes the expression of both AR and ARv7 target genes supporting its role as an AR and ARv7 coactivator.

To complement the above knockdown data, we tested the effects of inhibiting DNA-PK activity on AR/ARv7 target gene expression in LNCaP dox-inducible ARv7 cells. These cells were starved in CSS for 48 hours before being treated with NU7441 (two different doses) for 1 hour before being treated with R1881 or dox for 24 hours to induce AR and ARv7 activity, respectively. Inhibiting DNA-PK resulted in reduction of AR/ARv7-driven expression of *PSA* (Fig 6B, *top* panel) and *FKBP5* (Fig 6B, *middle* panel), and of the ARv7-specific target gene *EDN2* (Fig 6B, *bottom* panel). Collectively, the data in Fig 6 suggest that DNA-PK is a *bona fide* coactivator of both AR and ARv7 transcriptional activity, consistent with previous reports from the Knudsen [41,64] and Gaughan laboratories [35].

### Discussion

There is compelling evidence that ARv7 expression correlates with resistance to second generation ADT/ARSI [18,19] implicating ARv7 as a contributor to CRPC. Although ARv7 regulates many of the same gene targets as AR, several reports have identified genes that are preferentially regulated either by AR or ARv7 [39,42,67–70]. Differential binding to chromatin is one mechanism for preferential regulation, but differences in recruitment of CoRs likely also contributes to

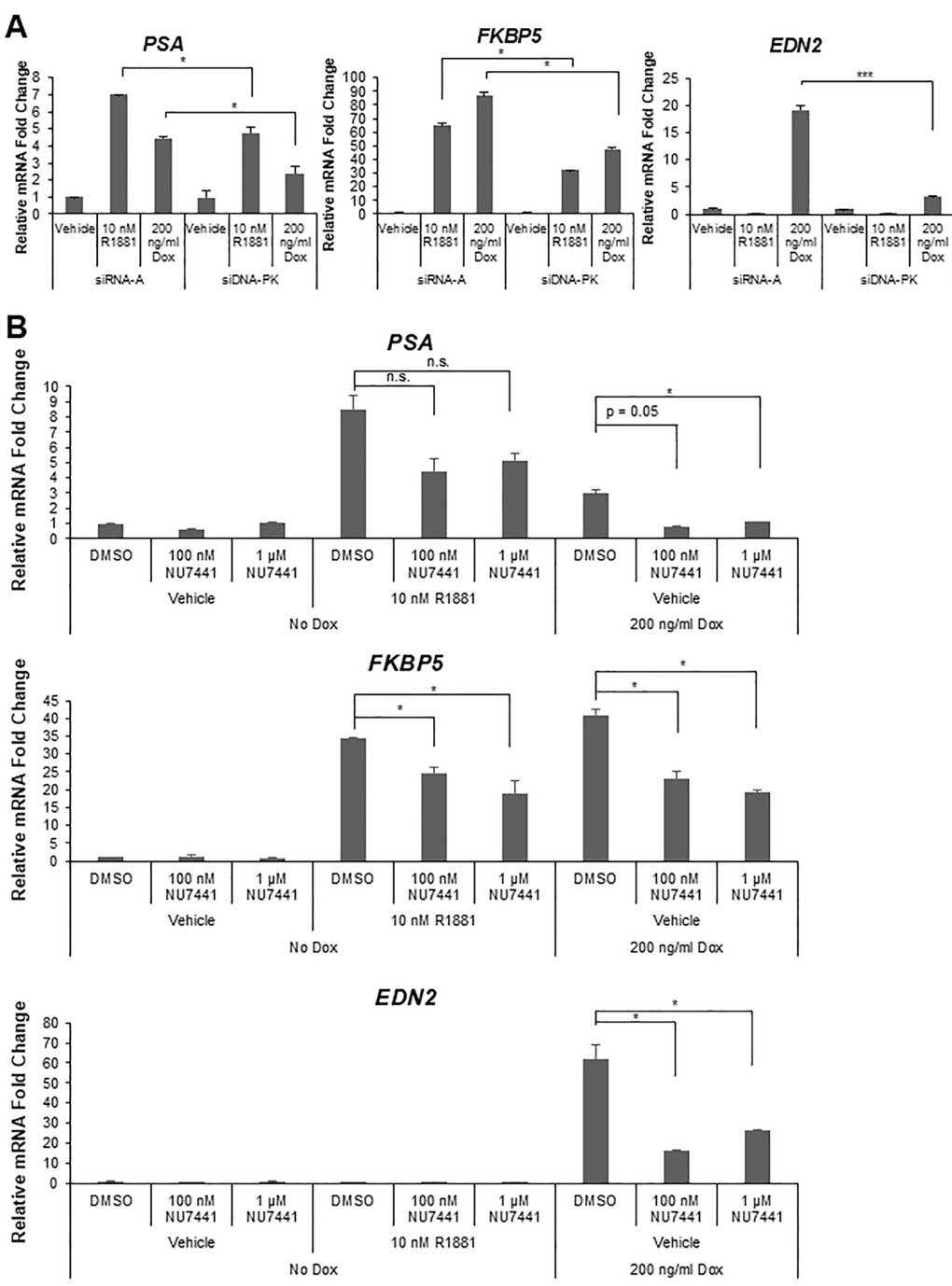

**Fig 6. DNA-PK coactivates AR and ARv7 target gene expression in LNCaP dox-inducible ARv7 cells. (A)** LNCaP dox-inducible ARv7 cells were grown in CSS serum and treated with siRNAs targeting DNA-PKcs for 48 hours. Cells were treated with R1881 (10 nM) or dox (200 ng/ml) for 24 hours. Shared AR/ARv7 targets (*PSA*, *FKBP5*) and an ARv7-selective target gene *EDN2* were reduced by loss of DNA-PK. P-values: for *PSA*+R1881, p=0.032; for *PSA*+dox, p=0.0164; for *FKBP5*+R1881, p=0.032, for *FKBP5*+dox, p=0.024; for *EDN2*+dox, p=0.00073. **(B)** LNCaP dox-inducible ARv7 cells grown in CSS serum were starved for 48 hours before being treated (-/+) NU7441 (100 nM or 1 μM) for 1 hour. Cells were then either treated with R1881 (10 nM) or dox (200 ng/ml) for 24 hours. Shared AR/ARv7 targets (*PSA*, *FKBP5*) and an ARv7-selective target gene *EDN2* were reduced by DNA-PK inhibition. In all panels, gene expression analyses were done via RT-qPCR and relative mRNA levels were determined after normalization to human *GAPDH* mRNA. P-values: for *PSA*+dox+100 nM NU7441, p=0.050; for *PSA*+dox+1 μM NU7441, p=0.018; for *FKBP5*+R1881+100 nM NU7441, p=0.041; for *FKBP5*+R1881+1 μM NU7441, p=0.017; for *FKBP5*+dox+100 nM NU7441, p=0.011; for *FKBP5*+dox+1 μM NU7441, p=0.013; for *EDN2*+dox+100 nM NU7441, p=0.026; for *EDN2*+dox+1 μM NU7441, p=0.039. *p<0.05, ***p<0.001, n.s.=not significant.

unique activities. Because binding to DNA can alter the structure of proteins, we sought to identify proteins that interact with ARv7 bound to DNA and to ask whether AR showed a similar capacity to interact with these proteins. To eliminate differences due to potential binding of other transcription factors, we developed an assay using a biotinylated DNA fragment containing a promoter with three upstream AREs, but no known consensus binding sites for other transcription factors. This template was sufficient to induce AR- or ARv7-dependent transcription in the presence of HeLa nuclear extract and purified receptor (Fig 1A). To optimize detection of associated proteins, levels of purified ARv7 used in the binding assay were titrated and the amount yielding the highest recruitment of a known AR coactivator, CARM1 (S1B Fig in S1 File), was used for an unbiased mass spectrometry analysis of proteins that interacted with ARv7 bound to DNA. Numerous proteins were identified (Fig 1C; see S5 Table for the full searchable list) and further validated by immunoblotting (Fig 1D; S1C and S1D Figs). Several novel proteins were identified. To test whether these proteins also bound to AR, the amount of AR that bound the highest levels of CARM1 was identified (S1B Fig in S1 File) and used to detect other proteins associated with AR. ARv7 was substantially more transcriptionally active than AR on our tested 3xARE template (Fig 1A) and more AR was required to optimally recruit CARM1 (S1B Fig). None of the candidates were uniquely specific for ARv7, but CARM1, CAND1, and ANP32A exhibited both more absolute binding to ARv7 and proportional binding relative to the amount of receptor (Fig 1E).

The potential coregulators with enhanced recruitment to ARv7 warranted further investigation of their functional relevance for AR and ARv7 activity. CAND1 was originally identified as TIP120, a TATA-binding protein (TBP)-interacting protein that stimulated RNA pol I, II, and III transcription [59] and associated with SUG1 in a ~ 800 kDa complex [71], but it was not reported as an AR coactivator. ANP32A was originally called pp32 and one group reported that it associated with AR and retinoblastoma protein [60]. However, neither protein was identified as being associated with ARv7. There are limited data published on their roles in AR/ARv7-driven transcription and this is why they became the focus of further study. Here, we filled this knowledge gap by showing that knockdown of ANP32A and CAND1 in LNCaP and LNCaP dox-inducible ARv7 cells significantly reduced both AR and ARv7 target gene expression (Fig 2; S2 and S3 Figs in S1 File). Moreover, CAND1 was recruited by both AR and ARv7 at two target genes (Fig 4), but the ratio of CAND1 occupancy to ARv7 was higher than that of CAND1 occupancy to AR as was observed in our *in vitro* assays (Fig 1E).

A previous study correlated higher *CAND1* mRNA and protein in prostate tumors versus normal tissue [72]. However, the clinical significance of these elevated expressions was not investigated. In this study, bioinformatic approaches investigating the clinical relevance of CAND1 in human prostate cancer sample datasets revealed that *CAND1* mRNA expression correlated with metastatic prostate cancer; high *CAND1* expression correlated with poor patient prognosis and an increased risk of recurrence (Fig 3). Our findings are consistent with a report that high *CAND1* mRNA levels were associated with higher tumor recurrence and decreased overall survival in prostate cancer patients [73]. The discovery of CAND1 as a coactivator for both AR and ARv7 implies it or its interaction surfaces could be novel therapeutic targets in CRPC.

DNA-PK has been reported to associate with AR and ARv7 to stimulate their transactivation [35,41], which we have further confirmed in cell-free transcription and cell-based assays by depleting DNA-PK using siRNA or inhibiting its activity with NU7441 (Fig 5F; Fig 6; S4B and S5 Figs in S1 File). However, its exact role(s) in the transcriptional activation process had not been identified. We found that DNA-PK activation induced phosphorylation of both AR and ARv7 as judged by phosphatase-sensitive gel mobility shifts that also were sensitive to DNA-PK depletion and inhibition (Fig 5). That the phosphorylation was a direct result of DNA-PK activation rather than secondary to another DNA-PK dependent activity was confirmed using purified DNA-PK (Fig 5D). Based on our previous study with ERα [40], it is likely that AR is one of several/ many proteins phosphorylated by DNA-PK that contribute to AR isoform activity. To complement our cell-free ARE DNA pull-down experiments, AR and ARv7-driven cell-free transcription was reduced by inhibition of DNA-PK (Fig 5F). We found that one component of the role of DNA-PK activity is to mediate enhanced stabilization of CAND1 and ANP32A association with AR and ARv7 bound to AREs (Fig 5E). These studies are consistent with our previous study of ERα-coregulator complexes

showing a role for DNA-PK in phosphorylating known activating serines of both ERα and SRC-3 and stabilizing coactivator binding in those complexes [40]. Given the data presented here and in a prior report [35] showing that ARv7 is very sensitive to DNA-PK inhibition, DNA-PK targeting may be a good therapeutic approach treating ARv7-expressing dominant CRPCs. A recent Phase 1b clinical trial (NCT02833883) of metastatic CRPC patients that have progressed on ADT tested the safety and efficacy of combining CC-115, a dual DNA-PK and mammalian target of rapamycin (mTOR), with enzalutamide [74]. While the combination was well tolerated, CC-115 was insufficient to inhibit DNA-PK but showed a trend to decreased PSA response in patients with PI3 kinase alternations. This study suggests that additional DNA-PK inhibitors should be tested for better efficacy in the CRPC patients, such as a new trial of peposertib in combination with radium-223 (NCT04071236).

ARv7 lacks exons 4–8 of AR and contains only a small amount of unique sequence, which to date has not been shown to have any function beyond presumably substituting for the nuclear localization signal found in exon 4. Thus, it was possible that ARv7 would interact with a subset of AR coregulators and not exhibit any preferential/unique interactions. However, three proteins, CAND1, CARM1, and ANP32 interacted more strongly with ARv7 as compared to AR in our biochemical, cell-free ARE pulldown assay (Fig 1E). Importantly, the enhanced interaction of CAND1 with ARv7 was confirmed in prostate cancer cells using ChIP assays (Fig 4). Our recent cryo-EM study [23] showed that the N-terminus of AR interacts with the C-terminus, consistent with earlier biochemical studies (e.g., [75]). Thus, the C-terminus of AR may be competing for interactions with CoRs contributing to the enhanced binding of some coactivators to ARv7 and its enhanced activity, at least, on promoters containing only AREs. Inhibition of AR isoforms lacking the C-terminal LBD has been challenging, although there are now different antagonists that bind the N-terminus and reduce prostate cancer growth in pre-clinical xenograft models (e.g., UT-143 [76], EPI-7170 [77], SC912 [78], and the PROTAC NP18 [79]). Our studies suggest that, in addition to approaches that reduce the expression of AR isoforms or inhibit isoform activity by binding the N-terminus, targeting identified coactivators or their interactions with AR and ARv7 should be effective in blocking activity and thus growth of AR isoform-dependent tumors.

## Supporting information

**S1 File. Supplementary Material.** This file contains S1-S4 Tables, Legends for S1-S5 Fig, and S1-S5 Fig.
(DOCX)

**S5 Table. List of all proteins identified by mass spectrometry in ARv7 ARE DNA pulldowns (searchable Excel file).**
(XLSX)

**S6 Fig. Raw, uncropped images of all immunoblots with respective assayed proteins labeled with arrows (separate PDF file).**
(PDF)

## Acknowledgments

The authors thank members of the Tissue Culture Core (Judy Roscoe, Javier Pacheco, and Cheryl Parker), the Mass Spectrometry Proteomics Core (Dr. Sung Yun Jung, Antrix Jain, and Bhoomi Bhatt), and the Monoclonal Antibody / Recombinant Protein Production Core (Kurt Christensen) for their technical assistance with cell culture, MS/proteomics analyses and PRIDE database submission, and recombinant protein purification. The authors thank the late Drs. Bert W. O'Malley and Ming-Jer Tsai and Dr. Sophia Tsai for critical comments on the manuscript.

## Author contributions

**Conceptualization:** Ross A. Hamilton, Nancy L. Weigel, Charles E. Foulds.

**Formal analysis:** Ross A. Hamilton, Basil Paul, Kimal Rajapakshe, Anil K. Panigrahi, Sandra L. Grimm, Cristian Coarfa, Anna Malovannaya, Nancy L. Weigel, David M. Lonard, Charles E. Foulds.

**Funding acquisition:** Ping Yi, Cristian Coarfa, Anna Malovannaya, Nancy L. Weigel, David M. Lonard.

**Investigation:** Ross A. Hamilton, Basil Paul, Anil K. Panigrahi.

**Project administration:** Charles E. Foulds.

**Resources:** Ping Yi, Anil K. Panigrahi, Nancy L. Weigel.

**Supervision:** Nancy L. Weigel, Charles E. Foulds.

**Visualization:** Ross A. Hamilton, Basil Paul, Kimal Rajapakshe, Sandra L. Grimm, Cristian Coarfa, Anna Malovannaya.

**Writing – original draft:** Ross A. Hamilton, Nancy L. Weigel, Charles E. Foulds.

**Writing – review & editing:** Ross A. Hamilton, Basil Paul, Ping Yi, Kimal Rajapakshe, Anil K. Panigrahi, Sandra L. Grimm, Cristian Coarfa, Anna Malovannaya, Nancy L. Weigel, David M. Lonard, Charles E. Foulds.

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
