## [Decision Letter · Decision Letter 0]

1 Apr 2026

PONE-D-26-09739Identification of CAND1 as a DNA-dependent protein kinase-regulated coactivator of androgen receptor and the ARv7 splice variant PLOS One

Dear Dr. Foulds,

Thank you for submitting your manuscript to PLOS ONE. After careful consideration, we feel that it has merit but does not fully meet PLOS ONE’s publication criteria as it currently stands. Therefore, we invite you to submit a revised version of the manuscript that addresses the points raised during the review process.

We look forward to receiving your revised manuscript.

Kind regards,

Zoran Culig

Academic Editor

PLOS One

Journal Requirements:

5. Thank you for stating in your Funding Statement:

S.L.G., K.R., and C.C. were partially supported by CPRIT grants RP200504 and RP210227, by NIH P30 shared resource grant CA125123, by NIEHS grants P30 ES030285 and P42 ES027725, and by NIH grants S10OD032185, U54CA274321, and R21DE032344.

6. Please expand the acronym “NIH” and “CPRIT” (as indicated in your financial disclosure) so that it states the name of your funders in full.

7. Thank you for stating the following in the Competing Interests/Financial Disclosure section:

C.E.F and D.M.L disclose other support from CoRegen, Inc. unrelated to the current study.

We note that one or more of the authors are employed by a commercial company: CoRegen, Inc.

Within your Competing Interests Statement, please confirm that this commercial affiliation does not alter your adherence to all PLOS ONE policies on sharing data and materials by including the following statement: ""This does not alter our adherence to  PLOS ONE policies on sharing data and materials.” (as detailed online in our guide for authors http://journals.plos.org/plosone/s/competing-interests) . If this adherence statement is not accurate and  there are restrictions on sharing of data and/or materials, please state these. Please note that we cannot proceed with consideration of your article until this information has been declared.

8. When completing the data availability statement of the submission form, you indicated that you will make your data available on acceptance. We strongly recommend all authors decide on a data sharing plan before acceptance, as the process can be lengthy and hold up publication timelines. Please note that, though access restrictions are acceptable now, your entire data will need to be made freely accessible if your manuscript is accepted for publication. This policy applies to all data except where public deposition would breach compliance with the protocol approved by your research ethics board. If you are unable to adhere to our open data policy, please kindly revise your statement to explain your reasoning and we will seek the editor's input on an exemption. Please be assured that, once you have provided your new statement, the assessment of your exemption will not hold up the peer review process.

Additional Editor Comments:

The authors have addressed important issue of coregulator action of full length and variant androgen receptor. There are some minor suggestions for revision.

Reviewers' comments:

Reviewer's Responses to Questions

**Comments to the Author**

1. Is the manuscript technically sound, and do the data support the conclusions?

Reviewer #1: Yes

2. Has the statistical analysis been performed appropriately and rigorously? 

Reviewer #1: Yes

3. Have the authors made all data underlying the findings in their manuscript fully available?

Reviewer #1: Yes

4. Is the manuscript presented in an intelligible fashion and written in standard English?

Reviewer #1: Yes

5. Review Comments to the Author

Reviewer #1: This is an elegant mechanistic study on novel co-factors of AR and ARv7 in prostate cancer. The experiments are well designed and conclusions are justified by the results.

Please see some minor points below:

Line 80, p4: please add more recent studies on AR interactome, using co-IP and MS. For example: PLoS One. 2024 Dec 13;19(12):e0309491. doi:10.1371/journal.pone.0309491. eCollection 2024.PMID: 39671399

Line 102 p5: typo (coactivator)

Line 132 p6: typo, profiling twice

Line 154 p7: where are the modifications highlighted? This should be clearly indicated.

Figure 1 A, bottom panel: it is not clear what the labels DMSO/Vehicle and DMSO/100nM R1881 refer too. This is not described in the main text either.

Lines 391-393, p16: The authors explain why they use NE from Hela which is essentially to pick a cell line without AR/ARv7 and a good source of RNA pol II machinery and CoRs. However, would the CoRs be less specific to prostate cancer than using a prostate cancer cell line without AR/ARv7, such as PC3 or DU154? This point should be at least discussed.

Figure 4 A: the scale for the bottom graphs is different from the top, which is understandable considering the amount of change. However, in the bottom panel changes are a fraction of those at the top. Is there any repercussion on the biological significance and interpretation of these results?

Figure 5B: Are the labels at the top correct? Should second lane be + ATP?

Line 719 p29: typo ‘TATA binding interacting protein’.

6. PLOS authors have the option to publish the peer review history of their article (what does this mean?). If published, this will include your full peer review and any attached files.

Reviewer #1: No

---

## [Author Response · Author response to Decision Letter 1]

8 Apr 2026

A. Response to Academic Editor’s Concerns:

>After thorough review, our manuscript meets PLOS ONE’s style requirements.

2. PLOS ONE now requires that authors provide the original uncropped and unadjusted images underlying all blot or gel results reported in a submission’s figures or Supporting Information files.

>We have added all raw, uncropped images of immunoblots as S6 Fig in the Supporting Information.

When you submit your revised manuscript, please ensure that your figures adhere fully to guidelines.

> We used PLOS’s free figure tool, NAAS, to prepare publication quality figures for our six Figures.

>Per PLOS, supporting figures and supporting tables do not follow the same requirements as tables and figures in the main body of your manuscript, because PLOS hosts them on servers that can handle a wider variety of file types than our published articles. Because of this, we did not use NAAS to check any supporting information files that are presented as .DOCX, .XLSX, or .PDF files.

In your cover letter, please note whether your blot/gel image data are in Supporting Information.

>This has been indicated in our cover letter.

3. PLOS requires an ORCID iD for the corresponding author in Editorial Manager on papers submitted after December 6th, 2016. Please ensure that you have an ORCID iD and that it is validated in Editorial Manager.

>I have validated my ORCID iD as “0000-0003-4908-1473” in Editorial Manager.

4. We note that the grant information you provided in the ‘Funding Information’ and ‘Financial Disclosure’ sections do not match. When you resubmit, please ensure that you provide the correct grant numbers for the awards you received for your study in the ‘Funding Information’ section.

>We apologize for the error.

We have now revised the financial disclosure to read:

“The study was funded by National Institutes of Health grants R01HD007857 and R01HD008188 (DML); P42ES027725 SubProject 7644 and U54CA274321 SubProject 9353 (CC); and P30CA125123 SubProject 8884 (AM); Cancer Prevention and Research Institute of Texas grant RP150648 Project 2 (NLW); Department of Defense Congressionally Directed Medical Research Program grants W81XWH-17-1-0236 (NLW) and W81XWH-21-1-0404 (PY). The above funders had no role in study design, data collection and analysis, decision to publish, or preparation of the manuscript. CoRegen, Inc. provided support in the form of salaries for authors [AKP, PY, DML, and CEF], but did not have any additional role in the study design, data collection and analysis, decision to publish, or preparation of the manuscript. The specific roles of these authors are articulated in the ‘author contributions’ section. There was no additional external funding received for this study.”

5. Thank you for stating in your Funding Statement:

S.L.G., K.R., and C.C. were partially supported by CPRIT grants RP200504 and RP210227, by NIH P30 shared resource grant CA125123, by NIEHS grants P30 ES030285 and P42 ES027725, and by NIH grants S10OD032185, U54CA274321, and R21DE032344.

> As noted above, we have now revised the financial disclosure to read:

“The study was funded by National Institutes of Health grants R01HD007857 and R01HD008188 (DML); P42ES027725 SubProject 7644 and U54CA274321 SubProject 9353 (CC); and P30CA125123 SubProject 8884 (AM); Cancer Prevention and Research Institute of Texas grant RP150648 Project 2 (NLW); Department of Defense Congressionally Directed Medical Research Program grants W81XWH-17-1-0236 (NLW) and W81XWH-21-1-0404 (PY). The above funders had no role in study design, data collection and analysis, decision to publish, or preparation of the manuscript. CoRegen, Inc. provided support in the form of salaries for authors [AKP, PY, DML, and CEF], but did not have any additional role in the study design, data collection and analysis, decision to publish, or preparation of the manuscript. The specific roles of these authors are articulated in the ‘author contributions’ section. There was no additional external funding received for this study.”

>We have removed previous funding sources either provided by BCM PIs that are not on this manuscript or that simply supported cores (Daniel Kraushaar-CPRIT RP200504, Dean Edwards-CPRIT RP210227, Vlad Sandulache- NIH R21DE032344, Cheryl Walker- NIH P30ES030285, and Susan Hilsenbeck- NIH S10OD032185).

6. Please expand the acronym “NIH” and “CPRIT” (as indicated in your financial disclosure) so that it states the name of your funders in full.

> As noted above, we have now revised the financial disclosure to read:

“The study was funded by National Institutes of Health grants R01HD007857 and R01HD008188 (DML); P42ES027725 SubProject 7644 and U54CA274321 SubProject 9353 (CC); and P30CA125123 SubProject 8884 (AM); Cancer Prevention and Research Institute of Texas grant RP150648 Project 2 (NLW); Department of Defense Congressionally Directed Medical Research Program grants W81XWH-17-1-0236 (NLW) and W81XWH-21-1-0404 (PY). The above funders had no role in study design, data collection and analysis, decision to publish, or preparation of the manuscript. CoRegen, Inc. provided support in the form of salaries for authors [AKP, PY, DML, and CEF], but did not have any additional role in the study design, data collection and analysis, decision to publish, or preparation of the manuscript. The specific roles of these authors are articulated in the ‘author contributions’ section. There was no additional external funding received for this study.”

>This information is listed in the cover letter.

7. Thank you for stating the following in the Competing Interests/Financial Disclosure section:

C.E.F and D.M.L disclose other support from CoRegen, Inc. unrelated to the current study.

We note that one or more of the authors are employed by a commercial company: CoRegen, Inc.

> We apologize for this oversight.

We have revised the Competing Interest Statement as:

“AKP, PY, DML, and CEF disclose salary support from CoRegen, Inc. unrelated to the current study. This does not alter our adherence to PLOS ONE policies on sharing data and materials. All other authors have nothing to disclose.”

>This information is listed in the cover letter.

> As noted above, we have now revised the financial disclosure to read:

“The study was funded by National Institutes of Health grants R01HD007857 and R01HD008188 (DML); P42ES027725 SubProject 7644 and U54CA274321 SubProject 9353 (CC); and P30CA125123 SubProject 8884 (AM); Cancer Prevention and Research Institute of Texas grant RP150648 Project 2 (NLW); Department of Defense Congressionally Directed Medical Research Program grants W81XWH-17-1-0236 (NLW) and W81XWH-21-1-0404 (PY). The above funders had no role in study design, data collection and analysis, decision to publish, or preparation of the manuscript. CoRegen, Inc. provided support in the form of salaries for authors [AKP, PY, DML, and CEF], but did not have any additional role in the study design, data collection and analysis, decision to publish, or preparation of the manuscript. The specific roles of these authors are articulated in the ‘author contributions’ section. There was no additional external funding received for this study.”

>As Funding Statement has been revised, we updated the Author Contributions section of the online submission form including “funding acquisition” by NLW, CC, and AM.

> We have revised the Competing Interest Statement as:

“AKP, PY, DML, and CEF disclose salary support from CoRegen, Inc. unrelated to the current study. This does not alter our adherence to PLOS ONE policies on sharing data and materials. All other authors have nothing to disclose.”

>Cover letter for the revision now includes both statements.

8. When completing the data availability statement of the submission form, you indicated that you will make your data available on acceptance. We strongly recommend all authors decide on a data sharing plan before acceptance, as the process can be lengthy and hold up publication timelines. Please note that, though access restrictions are acceptable now, your entire data will need to be made freely accessible if your manuscript is accepted for publication.

> The raw MS proteomics data for the the (-/+) ARv7 ARE pulldowns deposited to the ProteomeXchange Consortium via the PRIDE partner repository with the dataset identifier PXD021462 is now public. We have cited this on lines 327-329 of the revised manuscript and with Reference #52.

>This is listed in the cover letter.

9. If the reviewer comments include a recommendation to cite specific previously published works, please review and evaluate these publications to determine whether they are relevant and should be cited.

>We reviewed the one citation that Reviewer #1 requested. It is relevant and is now cited as Reference #33.

>We reviewed the reference list. No retracted papers were cited.

>We only have two changes:

• As noted above, we reviewed the one citation that Reviewer #1 requested. It is relevant and is now cited as Reference #33.

• We have cited as Reference #52 the following database as it has now been made public: “Bhatt B, Hamilton RA. Project Name: Proteomic analyses identify ANP32A and CAND1 as androgen receptor and splice variant coactivators, whose interactions are regulated by DNA-dependent protein kinase. ProteomeXchange Consortium via PRIDE. Accession: PXD021462 (Publication Date: April 6, 2026): http://www.ebi.ac.uk/pride.”

11. The authors have addressed important issue of coregulator action of full length and variant androgen receptor. There are some minor suggestions for revision.

>We have fully addressed each concern of the Academic Editor (above) and Reviewer #1 (below).

B. Response to Reviewer #1’s Concerns:

This is an elegant mechanistic study on novel co-factors of AR and ARv7 in prostate cancer. The experiments are well designed and conclusions are justified by the results.

>Thank you for your evaluation of our study. We appreciate the reviewer’s comments that have improved the manuscript.

Please see some minor points below:

1. Line 80, p4: please add more recent studies on AR interactome, using co-IP and MS. For example: PLoS One. 2024 Dec 13;19(12):e0309491. doi:10.1371/journal.pone.0309491. eCollection 2024.PMID: 39671399

>Thank you for this suggestion. As this PLoS One paper did AR IP-MS and found Ku70/Ku80 and PELP1 as known coregulators, we now cite it as new Reference #33.

2. Line 102 p5: typo (coactivator)

>This has been corrected.

3. Line 132 p6: typo, profiling twice

>This has been corrected.

4. Line 154 p7: where are the modifications highlighted? This should be clearly indicated.

>We apologize this was not clear. We have revised this section of the Methods to make clear what the modifications were.

5. Figure 1 A, bottom panel: it is not clear what the labels DMSO/Vehicle and DMSO/100nM R1881 refer too. This is not described in the main text either.

>We apologize this was not clear. We have made a revised Fig 1A in the revised submission and described the legends in the main text now.

6. Lines 391-393, p16: The authors explain why they use NE from Hela which is essentially to pick a cell line without AR/ARv7 and a good source of RNA pol II machinery and CoRs. However, would the CoRs be less specific to prostate cancer than using a prostate cancer cell line without AR/ARv7, such as PC3 or DU154? This point should be at least discussed.

>To address the reviewer’s concern, we state in the revised manuscript: “As HeLa cells are derived from a cervical carcinoma, some coregulators recruited to AR or ARv7 could be specific to this line and not a prostate cancer cell.” For some experimental support, however, we performed an ARE DNA pulldown with endogenous AR coming from LNCaP prostate cancer cell nuclear extract (-/+) two different R1881 agonist concentrations and observed similar coregulator recruitments as with HeLa cell nuclear extract programmed with recombinant AR and R1881 (presented as a new S1C Fig). We also have included the raw immunoblot data in a revised S6 Fig for new S1C Fig. These additions are presented in lines 402-407.

>We have not done any ARE DNA pulldowns from PC3 or DU-145 cells. HeLa cells were used only to identify potential coregulator candidates with subsequent functional validation in LNCaP prostate cells using siRNA knockdown of candidates testing for an effect on known AR/ARv7 target genes. We also performed bioinformatic analyses of patient prostate cancer samples for the validated new candidates.

7. Figure 4 A: the scale for the bottom graphs is different from the top, which is understandable considering the amount of change. However, in the bottom panel changes are a fraction of those at the top. Is there any repercussion on the biological significance

---

## [Editor Report · Decision Letter 1]

27 Apr 2026

Identification of CAND1 as a DNA-dependent protein kinase-regulated coactivator of androgen receptor and the ARv7 splice variant

PONE-D-26-09739R1

Dear Dr. Foulds,

We’re pleased to inform you that your manuscript has been judged scientifically suitable for publication and will be formally accepted for publication once it meets all outstanding technical requirements.

Kind regards,

Zoran Culig

Academic Editor

PLOS One

Additional Editor Comments (optional):

No further comments necessary.
---

## [Editor Report · Acceptance letter]

PONE-D-26-09739R1

PLOS One

Dear Dr. Foulds,

I'm pleased to inform you that your manuscript has been deemed suitable for publication in PLOS One. Congratulations! Your manuscript is now being handed over to our production team.

Kind regards,

on behalf of

Dr. Zoran Culig

Academic Editor

PLOS One